# From “Traditional” to “Trained” Immunity: Exploring the Novel Frontiers of Immunopathogenesis in the Progression of Metabolic Dysfunction-Associated Steatotic Liver Disease (MASLD)

**DOI:** 10.3390/biomedicines13082004

**Published:** 2025-08-18

**Authors:** Mario Romeo, Alessia Silvestrin, Giusy Senese, Fiammetta Di Nardo, Carmine Napolitano, Paolo Vaia, Annachiara Coppola, Pierluigi Federico, Marcello Dallio, Alessandro Federico

**Affiliations:** 1Hepatogastroenterology Division, Department of Precision Medicine, University of Campania Luigi Vanvitelli, 80138 Naples, Italy; mario.romeo@unicampania.it (M.R.); alessia.silvestrin@unicampania.it (A.S.); giusy.senese@unicampania.it (G.S.); fiammetta.dinardo@studenti.unicampania.it (F.D.N.); carmine.napolitano1@studenti.unicampania.it (C.N.); paolo.vaia@studenti.unicampania.it (P.V.); annachiara.coppola@unicampania.it (A.C.); alessandro.federico@unicampania.it (A.F.); 2Pharmaceutical Department, ASL NA3 Sud, Torre del Greco, 80059 Naples, Italy; federicopierluigi@libero.it

**Keywords:** trained immunity, oxidative stress, inflammation, immunometabolism

## Abstract

Metabolic dysfunction-associated steatotic liver disease (MASLD) has emerged as the most prevalent chronic hepatopathy and a leading precursor of hepatocellular carcinoma (HCC) worldwide. Initially attributed to insulin resistance (IR)-driven metabolic imbalance, recent insights highlight a multifactorial pathogenesis involving oxidative stress (OS), chronic inflammation, and immune dysregulation. The hepatic accumulation of free fatty acids (FFAs) initiates mitochondrial dysfunction and excessive reactive oxygen species (ROS) production, culminating in lipotoxic intermediates and mitochondrial DNA damage. These damage-associated molecular patterns (DAMPs), together with gut-derived pathogen-associated molecular patterns (PAMPs), activate innate immune cells and amplify cytokine-mediated inflammation. Kupffer cell activation further exacerbates OS, while ROS-induced transcriptional pathways perpetuate inflammatory gene expression. Traditional immunity refers to the well-established dichotomy of innate and adaptive immune responses, where innate immunity provides immediate but non-specific defense, and adaptive immunity offers long-lasting, antigen-specific protection. However, a paradigm shift has occurred with the recognition of trained immunity (TI)—an adaptive-like memory response within innate immune cells that enables enhanced responses upon re-exposure to stimuli. Following non-specific antigenic stimulation, TI induces durable epigenetic and metabolic reprogramming, leading to heightened inflammatory responses and altered functional phenotypes. These rewired cells acquire the capacity to produce lipid mediators, cytokines, and matrix-modifying enzymes, reinforcing hepatic inflammation and fibrogenesis. In this context, the concept of immunometabolism has gained prominence, linking metabolic rewiring with immune dysfunction. This literature review provides an up-to-date synthesis of emerging evidence on immunometabolism and trained immunity as pathogenic drivers in MASLD. We discuss their roles in the transition from hepatic steatosis to steatohepatitis, fibrosis, and cirrhosis, and explore their contribution to the initiation and progression of MASLD-related HCC. Understanding these processes may reveal novel immunometabolic targets for therapeutic intervention.

## 1. Background

### Global Burden of Metabolic Dysfunction-Associated Steatotic Liver Disease (MASLD)

Driven by sedentary lifestyles, overnutrition, and the global “pandemic” spreading of metabolic syndrome, metabolic dysfunction-associated steatotic liver disease (MASLD) is rapidly becoming a predominant hepatopathy in Western countries, emerging as a critical public health challenge worldwide [1]. MASLD currently affects nearly one in four individuals globally (an estimated prevalence of approximately 38% among adults worldwide) [1], with epidemiological projections indicating a dramatic rise in the disease burden exceeding 55% by 2040 [1]. Alarmingly, this condition is expected to become the leading cause of end-stage liver disease, liver transplantation, and hepatocellular carcinoma (HCC) worldwide [2]. MASLD encompasses a broad clinical and histological spectrum, ranging from simple steatosis (SS) to steatohepatitis (MASH), advanced fibrosis (AF), cirrhosis, and eventually HCC. Notably, HCC can arise even in non-AF scenarios, including SS [3]. This phenomenon strongly suggests the existence of distinct carcinogenic mechanisms, possibly independent of the classical fibrotic pathway, warranting a deeper exploration of alternative pathogenic routes [3].

Although the “hepatocentric theory” has deeply and progressively revolutionized the conception of dysmetabolic hepatic steatosis, supporting MASLD as a systemic disease where the liver represents the “central organ” with the various cardiometabolic comorbidities (obesity, type 2 diabetes, dyslipidemia, and arterial hypertension) constituting the relative physiopathological “satellites”, the multifactorial pathogenesis of this condition remains only partially elucidated [4].

While traditional theories have focused on insulin resistance (IR), lipotoxicity, oxidative stress (OS), and chronic inflammation as the main pathogenetic drivers, these factors alone no longer suffice to explain the complexity and variability of the clinical course [5]. In this sense, the incomplete identification of specific targets involved in only partially explored pathogenetic mechanisms contributing to the onset and progression of the disease reflects the current open challenge in managing MASLD patients [5].

In this scenario, where reliable biomarkers are still lacking, the accurate non-invasive diagnosis/staging of MASLD and the availability of non-invasive prognostic tools remain suboptimal, the implementation of effective personalized HCC surveillance strategies in non-AF stages is problematic, and pharmacological treatments have only recently been proposed [6,7,8].

A growing scientific consensus supports that the investigation of MASLD pathogenesis has reached a plateau in “classical” mechanistic understanding, and the enlargement of the currently pathogenetic scenario through the investigation of further novel perspectives represents an urgent need [9].

Considering this background, this literature review aims to explore the classic (“traditional” immunity) and novel (“trained” immunity) implications of immune mechanisms in hepatic steatosis progression, providing a state-of-the-art overview of the most recent evidence investigating the role of this emerging driver in the worsening of MASLD toward MASH, fibrosis, and cirrhosis, as well as its implications in MASLD-related tumorigenesis.

## 2. From “Classic” to “Untraditional” MASLD Pathogenesis

For decades, the scenario of MASLD pathogenesis has been primarily dominated by IR and relative dysmetabolic sequelae [5,9,10]. However, accumulating evidence has progressively underscored a multifaceted interplay involving oxidative stress (OS), chronic inflammation, and gut dysbiosis, ultimately determining immune dysregulation, promoting disease progression [4].

### 2.1. IR, Inflammation, OS, and Gut Dysbiosis as Triggers of Hepatic Immune Dysfunction

Lipid droplets (LDs), once regarded as inert reservoirs for neutral lipids in hepatic steatosis, have been recently recognized as dynamic organelles that play a central role in cellular metabolism and immune regulation [4]. In the context of MASLD, the excessive accumulation of LDs within hepatocytes reflects a state of metabolic imbalance, primarily driven by IR and dysregulated lipid homeostasis [4]. This lipid overload induces lipotoxic stress, which compromises mitochondrial integrity and promotes the generation of reactive oxygen species (ROS) [4]. Thus, the IR-guided hepatic accumulation of free fatty acids (FFAs) initiates mitochondrial dysfunction, resulting in the excessive production of ROS, which promotes the generation of lipotoxic intermediates and mitochondrial DNA damage [4]. These alterations give rise to damage-associated molecular patterns (DAMPs) that activate hepatic innate immune cells and stimulate the release of pro-inflammatory cytokines via activating NOD-like receptors (NLRs)-related pathways [including NLRP3-inflammasome] [4]. This cascade contributes to creating a pro-inflammatory hepatic microenvironment, wherein local hepatic immune cell activation further amplifies OS. In turn, ROS represent DAMPs and the aberrant ROS production activates ROS-sensitive transcription factors modulating the expression of inflammatory genes, perpetuating immune cell activation and contributing to fibrogenic remodeling [4].

Therefore, in this context, chronic inflammation and OS appear closely linked processes that play a central role in disease progression [11,12]

In recent years, the gut microbiota has progressively emerged as an important deus ex machina, crucially influencing these pathogenic processes [13]. Alterations in the gut microbiota composition and functioning (“gut dysbiosis”), indeed, significantly influence both inflammation and OS in the context of both MASLD and MASLD-related HCC [13]. The gut microbiota modulate the immune system through several microbe-derived metabolites [including short-chain fatty acids (SCFAs) and bile acids (BAs)], which in turn regulate systemic inflammation and intestinal integrity [13]. In the case of dysbiosis, on one side, the impairment of these mechanisms contributes to promoting a chronic inflammatory state that favors liver disease progression [14,15,16]. On the other hand, dysbiosis compromises the intestinal barrier integrity, leading to the translocation of different microbe-derived products [including lipopolysaccharides (LPS) and other potent inflammatory molecules] into the liver via the portal bloodstream (i.e., endotoxemia), locally acting as pathogen-associated molecular patterns (PAMPs). At the hepatic level, this event culminates in activating hepatic toll-like receptor (TLR)-related pathways, ultimately contributing to sustaining a microenvironment where hepatic inflammation fuels MASLD progression, fostering fibrosis and, ultimately, promoting HCC [12,15].

In addition to its metabolic and immunomodulatory roles, the gut microbiota profoundly influence hepatic oxidative stress homeostasis. Certain commensal microorganisms generate metabolites—including SCFAs and indoles—that exert antioxidant or redox-regulating effects within the liver. Intestinal dysbiosis alters this delicate equilibrium, facilitating ROS overproduction, mitochondrial dysfunction, and hepatocyte injury. Through the gut–liver axis, the microbiota functions as a critical interorgan communicator, shaping the hepatic microenvironment and contributing to the pathophysiological continuum from steatosis to hepatocarcinogenesis [11,14,17].

In this sense, the HCC arising in the setting of MASLD represents the extreme of a pathobiological continuum in which chronic inflammation and OS constantly interact [18]. In such a context, microbiota-derived factors remain critical: endotoxemia promotes angiogenesis, whereas altered BA pools and diminished SCFA production favor the expansion of immunosuppressive regulatory T cells and myeloid-derived suppressor cells, further weakening the anti-tumor defences [19,20].

The net result is a chronically inflamed yet immunotolerant hepatic microenvironment in which fibrotic scarring, oxidative DNA damage, and metabolic stress converge to create a fertile ground for malignant transformation [4,13,21].

Collectively, these findings underscore that immune dysregulation is not a secondary epiphenomenon, but rather a central determinant in the carcinogenic trajectory of MASLD.

They reinforce the concept that the liver, beyond its metabolic and detoxifying roles, functions as a highly immunologically active organ. Its distinctive immune architecture—characterized by a finely tuned balance between tolerance and rapid inflammatory reactivity—can be subverted under pathological conditions, facilitating immune evasion and malignant transformation. In this context, the liver emerges as a bona fide immune organ: a great proportion (~70%) of its blood supply arrives from the gut, continuously delivering microbial antigens and metabolites that are surveyed by a dense network of local parenchymal and non-parenchymal innate immune cells [22].

This unique architecture imposes a delicate balance between immune tolerance—required to prevent unwanted reactions to commensal antigens—and rapid pro-inflammatory defence against pathogens or tissue injury (“traditional” immunity). In recent decades, this last feature, considering also the relative pathogenetic implications, has fueled research efforts to furtherly and deeply investigate the mechanisms regulating the innate immune response, opening the doors to the progressive affirmation of the novel immunological conception of “trained” immunity (see next subparagraph) [23].

### 2.2. “Trained” Immunity as Revolutionary Immunologic Pathogenetic Frontier

In recent years, research efforts have shifted to the identification of novel and untraditional pathogenetic paradigms, suggesting that immunological mechanisms play a pivotal role in MASLD progression [4]. In parallel, the emerging theory of trained immunity (TI), supporting the existence of immune memory that confers long-term functional reprogramming even of innate immune cells, determining relevant metabolic repercussions (“immunometabolism”), has emerged as a promising explanatory and revolutionary model [24].

Recent advances in immunology have challenged the classical dichotomy between innate and adaptive immunity, revealing that innate immune cells can undergo long-term functional reprogramming [24]. This process is triggered by exogenous antigenic stimuli, such as microbial components (e.g., β-glucan, LPS, or Bacillus Calmette–Guérin—the BCG vaccine), as well as endogenous danger signals (including oxidized lipids, uric acid, and the heme group) [24,25]. These stimuli induce epigenetic and metabolic rewiring in innate cells, resulting in a heightened and nonspecific inflammatory response upon subsequent exposures [24]. Relevantly, unlike adaptive immunity, TI does not rely on antigen specificity or clonal expansion, yet it can persist for extended periods and influence the systemic immune tone [26]. Therefore, the impact of TI in chronic inflammatory disorders appears crucial, including in hepatobiliary conditions, where persistent local innate activation may significantly contribute to disease progression [27]. Relevantly, the implications of TI appear particularly compelling considering that, as previously highlighted, the liver represents the first organ downstream from the gut, heavily influenced by several microbiota-derived stimuli, as well as an immunologically active site, playing a central role in systemic inflammation and OS [28].

In the specific context of MASLD, the implications of TI appear predominantly deleterious [27]. Chronic exposure to metabolic stressors—including elevated levels of FFAs, lipotoxic intermediates, and gut-derived PAMPs—can induce a trained phenotype in hepatic innate immune cells [27].

These reprogrammed cells exhibit a heightened pro-inflammatory and pro-fibrotic profile, characterized by the increased production of cytokines, chemokines, and matrix-remodeling enzymes [27].

Such responses contribute to the perpetuation of hepatic inflammation, OS, and extracellular matrix deposition, thereby facilitating the progression from SS to MASH, fibrosis, and, ultimately, HCC [27].

Therefore, while TI may confer protective effects in acute infectious settings [24,26], its sustained activation in MASLD appears to exacerbate the disease pathology.

The maladaptive nature of TI in this context underscores the need for therapeutic strategies aimed at modulating innate immune memory and its metabolic underpinnings.

Table 1 compares the “traditional immunity” with “trained immunity” response, reporting the most relevant differences, as well as the crucial specific potential implications in MASLD pathogenesis (Table 1).

## 3. “Traditional” Immunity in the Progression of MASLD

### 3.1. Liver as an Immunological Organ

The liver is a highly specialized and multifunctional organ that plays a central role in regulating diverse physiological processes, serving as the principal site for the synthesis, metabolism, and catabolism of macronutrients—including proteins, carbohydrates, and lipids—as well as the activation and storage of essential vitamins [33]. Additionally, the liver is essential for the biotransformation and detoxification of hormones, drugs, and other substances [34].

In recent years, besides these classic metabolic regulation activities, hepatic parenchyma has been increasingly recognized as an immunological site, and the liver, as it influences both innate and adaptive immune responses, has been proposed as a proper immunological organ [35]. Furthermore, the liver plays a key role in producing hepatokines, acute-phase proteins, and complement components, also influencing several extra-hepatic physiological and pathological processes [36].

In this sense, this organ is capable of identifying, capturing, and eliminating pathogens and foreign antigens that enter the bloodstream, thanks to a highly specialized network of liver-adapted immune cells [35]. The liver, indeed, is populated by a repertoire of specialized immune-related cells [including liver sinusoidal endothelial cells (LSECs), intrahepatic macrophages (KCs), and hepatic dendritic cells (DCs)], which possess antigen-presenting capabilities and actively participate in immune surveillance and activation. These cells form a dynamic interface between the bloodstream and hepatic parenchyma, contributing to the organ’s tolerogenic yet responsive immunological environment [37]. Below, the various types of immune cells populating the liver are rapidly reviewed.

#### 3.1.1. Liver Sinusoidal Endothelial Cells (LSECs)

LSECs are highly permeable and excel in endocytosis via mannose receptors (MR), scavenger receptors (SR), and Fc-gamma (Fc-γ) receptors, aiding in waste clearance [38]. In particular, LSECs are capable of both presenting and cross-presenting antigens to adaptive immune cells (CD4^+^ and CD8^+^ T cells) [38]. This unique ability allows them to prime naïve T cells and induce either immune activation or tolerance depending on the context. In this sense, due to constant exposure to gut-derived antigens via portal blood, LSECs promote immune tolerance by inducing regulatory T cells and suppressing excessive immune activation [38]. This regulation is crucial for preventing autoimmunity and maintaining liver function. Collectively, LSECs act as sentinel cells, detecting PAMPs and clearing circulating antigens and debris through endocytosis, contributing to maintaining innate immunity response homeostasis [38].

#### 3.1.2. Kupffer Cells (KCs)

Kupffer cells represent the liver’s resident macrophages, with a key role in microbial clearance from the portal vein, maintaining liver homeostasis. Located in hepatic sinusoids, they interact with sinusoidal endothelial cells, hepatic stellate cells (HSCs), and natural killer (NK) cells, enabling rapid immune responses to gut-derived microorganisms [39,40].

KCs include distinct subsets (CD14^+^CD16^−^, CD14^+^CD16^+^, and CD16^+^ cells) and exhibit phenotypic plasticity. Depending on microenvironmental cues, they polarize into pro-inflammatory M1 macrophages [driven by LPS and Th1 cytokines, releasing interleukin (IL)-1β, IL-6, IL-12, IL-23, and tumor necrosis factor (TNF)-a] or anti-inflammatory M2 macrophages [induced by Th2 cytokines IL-4 and IL-13, producing IL-10 and Transforming growth factor (TGF)-β] [41,42].

#### 3.1.3. Hepatic Stellate Cells (HSCs)

Also known as “Ito cells”, these cells can transition into a proliferative myofibroblast-like state in response to oxidative stress and inflammation, marked by the upregulation of α-smooth muscle actin (αSMA) [43]. They play key roles in liver regeneration, sinusoidal tension modulation, and blood flow regulation, while also contributing to immunomodulation [44,45].

#### 3.1.4. Dendritic Cells (DCs)

The liver’s antigen-presenting cell population includes DCs, which are less abundant than in other organs. Primarily located in the peripheral veins and space of Disse, hepatic DC activation requires FMS-like tyrosine kinase 3 ligand (FLT3L) and granulocyte–macrophage colony-stimulating factor (GM-CSF) [46,47]. Hepatic DCs are classified into myeloid (mDCs) and plasmacytoid (pDCs) subsets [46,47]. About one-third of hepatic CD11c mDCs express CD141, a marker found in less than 5% of circulating mDCs. Liver DCs exhibit an immature phenotype with low MHC-II expression and minimal co-stimulatory molecules (CD40, CD80, and CD86) [48].

#### 3.1.5. Natural Killer Cells (NK Cells) and Natural Killer T Cells (NKT Cells)

Hepatic NK cells, key effector lymphocytes of the innate immune system, comprise two subsets: liver-resident NK cells (LrNK, or innate-like T cells “ILC1”), localized in hepatic sinusoids, and circulating classical NK cells (cNK), resembling those in peripheral blood and the spleen [49,50]. They differ from peripheral NK cells in surface markers, cytokine profiles, and cytotoxicity [49]. The liver is also enriched in NKT cells, primarily classified into type I (invariant, iNKT) and type II subsets. iNKT cells express semi-invariant T cell receptors (TCR) (Vα24-Jα18), while type II NKT cells have a more diverse TCR repertoire [51]. Through rapid cytokine secretion and direct interactions with hepatocytes, KCs, and DCs, they bridge innate and adaptive immunity. iNKT cells contribute to immune surveillance, inflammation control, and anti-tumor defense; conversely, their dysregulation may promote liver injury and fibrosis [51].

Collectively, this evidence suggests that liver functions are essential for maintaining systemic homeostasis through complex, interdependent networks and signaling pathways that often involve interactions among different cell types [37]. Hepatic injury (e.g., viral infection-related damages) can compromise the hepatic immune (metabolic) functions, impairing its ability to neutralize harmful stimuli and resolve inflammation. This disruption of immune tolerance and tissue integrity contributes to the breakdown of homeostatic mechanisms, thereby predisposing individuals to chronic inflammatory conditions, autoimmune phenomena, and hepatic (or extrahepatic) tumorigenesis [36].

### 3.2. “Traditional” Immunity Dysregulation in Driving the MASLD Progression

As MASLD progresses from SS to MASH and, ultimately, to AF, a pronounced escalation in hepatic immune cell infiltration has been reported. This inflammatory milieu is tightly regulated by immune activation, predominantly mediated by monocytes and macrophages [52].

Anyway, besides these, in this complex scenario, several innate immune cell types become activated, leading to the overproduction of pro-inflammatory cytokines. This cytokine release amplifies the hepatic recruitment of immune cells and exacerbates inflammation and hepatocellular injury [53]. Moreover, a substantial remodeling of the hepatic immune architecture, including the depletion and phenotypic reprogramming of resident KCs, the recruitment of monocyte-derived macrophages (MDMs), and the emergence of specialized MASH-associated macrophages (MAMs), is observed [54].

Finally, a critical transitional event driving the progression from MASLD to MASH is the interplay between innate and adaptive immune systems, wherein adaptive immune cells not only become activated, but also orchestrate the behavior of innate effectors, thereby reshaping the inflammatory landscape [55,56].

Considering this, to better understand the immunopathogenesis of MASLD progression, it is essential to characterize the key immune cell populations involved, including both innate and adaptive immune components that contribute to liver inflammation.

#### 3.2.1. Role of Macrophages: Expanding the Classic “Polarization Paradigm”

Macrophages play a pivotal role in the transition from SS to MASH, acting as central orchestrators of hepatic inflammation, fibrogenesis, and cancerogenesis [4]. Two main subsets have been identified as crucial protagonists driving disease worsening: resident KCs and circulating MDMs [56,57].

In the early stages of disease, metabolic stress (including lipotoxic stimuli, FFAs, and cholesterol crystals) and PAMPs (including LPS) activate KCs via TLR4/Nuclear Factor-κb (NF-κB) pathways and promote a proinflammatory mediator release [53]. Therefore, resident KCs undergo phenotypic reprogramming (“M1 polarization”) in response to toxic stimuli, increasing the release of pro-inflammatory cytokines [TNF-α, IL-1β, and CC Motif Chemokine Ligand 2 (CCL2)] [53].

This inflammatory cascade promotes the recruitment of MDMs, which further amplify hepatic injury and contribute to the emergence and activation of specialized subsets (including, among others, MAMs) [54]. This shift replaces embryonically derived KCs with MDMs, accelerating disease progression [54]. In support of this, evidence has demonstrated that cholesterol-loaded macrophages exhibit enhanced M1 polarization and inflammasome activation, particularly via the NLRP3–IL-1β axis, exacerbating hepatocyte damage [58]. Ganz et al. previously reported that prolonged exposure to a high-fat (HFD), high-cholesterol, high-sugar diet in mice leads to cumulative DAMP accumulation, inflammasome activation, and macrophage-driven fibrosis [59]. More recently, Li et al. showed that dietary cholesterol suppresses 7-Dehydrocholesterol Reductase (DHCR7) expression in hepatic macrophages, promoting a pro-inflammatory phenotype (M1) and worsening steatohepatitis in HFD-fed mice, an effect reversible by simvastatin treatment [58].

Beyond their original/embryogenic heterogeneity, resident hepatic macrophages display diverse phenotypes, driving hepatic inflammation via cytokines and ROS, or contributing to fibrosis during tissue repair [60]. Concerning this, Zhang et al. recently showed in a mouse model that YAP activation via mammalian Sterile 20-like kinase 1 and 2 (Mst1/2) deletion enhances both inflammatory and reparative functions of Kupffer-1 and -2 cells (KC1 and KC2), partly through C1q, increasing inflammation and fibrosis. These findings suggest that YAP inhibition may offer a novel therapeutic avenue in MASH [57]. Additionally, Zou et al. identified macrophagic miR-204-3p as a key regulator of MASH. The authors reported a significant correlation between reduced miR-204-3p levels and disease severity, enhancing inflammation and lipid accumulation [61]. Mechanistically, miR-204-3p was shown to inhibit TLR4/JNK signaling and promote autophagy by activating ULK1 transcription and the VPS34 complex. These data reveal a novel nuclear function of miR-204-3p and support therapeutic potential in MASLD [61].

Altogether, these findings underscore the multifaceted role of macrophages as both initiators and perpetuators of hepatic inflammation, positioning them as promising targets for therapeutic intervention in MASLD.

Furthermore, recent advances in single-cell transcriptomic profiling have revolutionized our understanding of hepatic macrophage heterogeneity in MASLD, revealing a spectrum of distinct subpopulations with specialized spatial and functional attributes.

These findings challenge the traditional binary M1/M2 polarization paradigm, highlighting instead a dynamic continuum of macrophage phenotypes that evolve in response to metabolic and inflammatory cues. For instance, Blériot et al. identified transcriptionally distinct macrophage subsets in murine models of MASLD, including lipid-associated macrophages (LAMs), which exhibited immunosuppressive features [62].

Similarly, Ramachandran et al. demonstrated that MDMs infiltrating the steatotic liver acquire context-dependent transcriptional programs that differ markedly from resident KCs, contributing to disease progression through cytokine secretion and extracellular matrix remodeling [63]. These insights have been corroborated by spatial transcriptomic analyses, which reveal that macrophage localization within hepatic lobules—particularly in proximity to injured hepatocytes and activated HSCs—correlates with their pro-inflammatory and fibrogenic potential [42,64]. Collectively, these studies underscore the importance of macrophage plasticity and spatial dynamics in MASLD pathogenesis and suggest that strategies targeting specific macrophage subsets may offer greater precision than therapeutic approaches based on classical polarization states.

#### 3.2.2. Role of Neutrophils: A Biface Janus Mediating Liver Injury and Repair

Neutrophils have emerged as critical mediators in the transition from hepatic SS to MASH, contributing to both the initiation and amplification of liver inflammation [65]. These cells are recruited via chemokines like CXCL1 and IL-8 (CXCL2) and adhere to LSECs through ICAM-1 and integrin αMβ2 (Mac1), in response to IL-1β from KCs [38,65]. In the MASLD context, indeed, upon exposure to lipotoxic stimuli, KCs and damaged hepatocytes release chemokines (such as CXCL1 and IL-8), which recruit neutrophils to the liver via CXCR2-dependent signaling [38,65]. Upon migration, neutrophils respond to DAMPs and complement activation independently of CXCL1/2 gradients [66], releasing cytokines, ROS, proteases, and forming neutrophil extracellular traps (NETs)—structures that influence inflammation, injury, and regeneration [65].

In this setting, NETs, triggered by a hepatic lipid overload, intensify inflammation by activating macrophages and shaping T cell responses, inducing Treg polarization and CD8^+^ T cell exhaustion via PD-L1 [67,68,69]. Concerning this, recent evidence has highlighted the pivotal role of sphingosine-1-phosphate receptor 2 (S1PR2) signaling in modulating neutrophil behavior during the early stages of liver injury. The activation of S1PR2 by sphingosine-1-phosphate (S1P), a bioactive lipid mediator, can redirect the neutrophil fate from apoptosis toward NETosis, a specialized form of cell death characterized by the release of NETs. These chromatin-based structures, enriched with histones and granule proteins such as myeloperoxidase and neutrophil elastase, contribute to fibrin deposition and the amplification of sterile inflammation. In murine models of fatty liver disease and cholestatic injury, S1PR2-mediated NETosis has been shown to exacerbate hepatic damage by promoting macrophage recruitment, polarization toward a pro-inflammatory phenotype, and the release of cytokines such as IL-1β and TNF-α. Moreover, NET-derived DAMPs further stimulate innate immune responses, creating a feed-forward loop that sustains neutrophil and macrophage infiltration [67,68,70]. In addition, the pharmacological inhibition or genetic silencing of S1PR2 has been demonstrated to attenuate NET formation, reduce fibrin accumulation, and mitigate liver inflammation and fibrosis, underscoring its potential as a therapeutic target in early-stage MASLD [67,68,70].

Moreover, neutrophil effectors such as elastase (NE), myeloperoxidase (MPO), and human neutrophil peptides (HNP-1) have been shown to exacerbate liver inflammation, IR, steatosis, and fibrosis in MASLD [71,72]. Relevantly, NE enhances KC activation and lipid dysregulation, while its absence mitigates early MASH features [71,72]. In support of this, Zang et al. initially demonstrated that neutrophil depletion via anti-Ly6G antibodies significantly attenuated liver inflammation and reduced serum alanine aminotransferase (ALT) levels in mice fed a methionine/choline-deficient (MCD) diet, implicating NE as a key effector in MASH pathogenesis [73]. Similarly, Zhou et al. subsequently revealed that neutrophil-HSC–HSC interactions prolong neutrophil survival and activate HSCs via GM-CSF and IL-15, establishing a feed-forward loop that promotes fibrosis in an HFD plus binge ethanol model [74]. More recently, transcriptomic analyses by Maretti-Mira et al. further showed that circulating neutrophils in advanced MASH patients exhibit a pro-inflammatory profile and extended lifespan, suggesting systemic priming before hepatic infiltration [75]. Collectively, these findings underscore the multifaceted role of neutrophils in MASLD progression and highlight their potential as therapeutic targets to intercept the inflammatory-to-fibrotic transition.

Anyway, neutrophils seem to exert dual roles in MASLD progression, contributing to both liver damage and repair. In this sense, these cells exhibit a paradoxical role in the pathogenesis of MASLD, functioning as both drivers of hepatic injury and mediators of tissue repair. While, as previously described, their infiltration into the liver promotes inflammation through the release of ROS, NE, and NETs, emerging evidence highlights their resolutive capacity under specific conditions. A key mechanism underpinning this duality involves neutrophil-derived microRNA-223 (miR-223), which modulates hepatic immune responses by silencing the NLRP3 inflammasome in macrophages and promoting their transition toward a restorative phenotype [76]. Neutrophil-derived miR-223 limits MASLD progression via immune modulation, and its loss has been shown to worsen liver injury (and promote hepatocarcinogenesis) [76].

In support of this, in murine models of spontaneous liver inflammation resolution, Calvente et al. demonstrated that the depletion of neutrophils or genetic ablation of miR-223 impairs inflammation resolution, exacerbates fibrosis, and accelerates hepatocarcinogenesis—effects reversible by the reconstitution of miR-223 or adoptive transfer of wild-type neutrophils [77]. These findings underscore the context-dependent immunoregulatory functions of neutrophils, positioning miR-223 as a critical molecular rheostat in the progression and resolution of MASLD.

Collectively, this controversial evidence suggests the need for further investigation to elucidate the net implications of neutrophil activation in the MASLD context.

#### 3.2.3. Hepatic Dendritic Cells in MASLD to MASH Progression: Limited Evidence

Hepatic DCs, comprising plasmacytoid (pDCs; PDCA-1^+^) and myeloid/classical DCs (mDCs; PDCA-1^−^), display functional heterogeneity [56]. In healthy livers, DCs are immature and poorly immunogenic. In MASLD, however, lipid accumulation—via the peroxisome proliferator-activated receptor gamma (PPARg) pathway—drives conventional DCs (cDCs) toward a pro-inflammatory, antigen-presenting state. Notably, suppressing lipogenesis can restore their tolerogenic function, highlighting the lipid content as a key regulator [46,56], and, relevantly, the activation of the liver kinase B1 (LKB1)-activated protein kinase (AMPK) /salt-inducible kinase (SIK) pathway has also been reported to allow DCs to inhibit Th17 cell differentiation, protecting against disease progression [78]. Another study by Song et al. showed that c-kit^+^ cDC1 cells (CD103^+/−^), which increase in MASLD but decline in MASH, exert anti-inflammatory effects. Their adoptive transfer in MASH mice reduced liver damage, likely via IL-10-mediated mechanisms [79]. Therefore, since hepatic DCs display functional heterogeneity that varies with the stage of MASLD, their precise contribution to disease pathogenesis remains to be fully elucidated [80].

#### 3.2.4. Natural Killer Cells and Natural Killer T Cells in MASLD to MASH Progression

NK cells, including conventional NK cells and liver-resident ILC1s, exhibit phenotypic similarities with functional differences that complicate their role in MASLD. NK cells are typically more cytotoxic, while ILC1s can acquire cytotoxicity in specific contexts [81,82,83]. In MASLD, the accumulation of lipids and oxidative stress-related damage products leads to altered NK cell subset distribution and an impaired effector function [84].

Notably, NK cells in MASLD transition from cytotoxic mediators to producers of pro-inflammatory cytokines (such as TNF-α and IFNγ), fueling local inflammation [84].

As MASLD advances to MASH, the NK cell compartment undergoes profound phenotypic reprogramming, marked by the reduced expression of activating receptors (e.g., Natural Killer group 2D—NKG2D-) and increased exhaustion markers (e.g., PD-1). These changes compromise their ability to lyse pathogenic targets and limit fibrogenic responses [84]. In MASH patients, increased circulating NK cells and the hepatic expression of NKG2D suggest dysregulated immune activation [84]. Interestingly, obesity has been shown to shift NK cells toward an ILC1-like phenotype with reduced cytotoxicity, potentially promoting inflammation in MASH while offering protection against HCC, depending on the disease stage [56,85,86].

Natural killer T (NKT) cells have been implicated in aggravating liver injury in models of obesity and alcohol-induced damage [87,88]. Moreover, these cells interact dynamically with other hepatic immune populations—such as KCs and infiltrating monocytes—amplifying inflammatory cascades and promoting hepatocyte injury in certain contexts [89]. Particularly, as similarly reported for neutrophils, in the context of MASLD, these cells exhibit dual behavior—initially contributing to the immunosurveillance and control of HSC activation, yet progressively succumbing to the deleterious effects of lipotoxicity and chronic metabolic stress [89]. Concerning this, relevant evidence has further evidenced, along with the advanced stages’ progression of MASH, an accumulation of NKT cells in the liver, which can drive fibrosis by activating HSCs [89]. In this sense, conflicting data contrarily support a protective role of iNKT cells. Regarding this, an NKT deficiency has been associated with fibrosis worsening in HFD-fed mice, especially in Th2-biased models [90].

Collectively, these findings support the role of NK and NKT cells as a crucial arm of innate immunity with increasingly appreciated roles in hepatic homeostasis and MASLD disease progression, whose definitive role has to be further investigated, as well as the cruciality of innate–adaptive immune cross-talk in influencing the immune response in dysmetabolic scenarios.

#### 3.2.5. The Emerging Role of Adaptive Immune Cells in MASLD to MASH Progression

Although the predominant role has historically been highlighted in viral-related chronic liver disorders, adaptive immunity, mediated by diverse subsets of T (CD4^+^, CD8^+^, γδ, and Treg) and B cells (transitional, memory, regulatory, etc.), is increasingly recognized as a key player also in MASLD and its progression to MASH [90,91,92,93].

CD4^+^ T cells exhibit functional plasticity, contributing to either pro- or anti-inflammatory responses via distinct subsets—Th1, Th2, Th17, and Treg—characterized by specific cytokines and transcription factors [94,95]. In MASLD, while data on the total hepatic CD4^+^ T cell numbers are conflicting [96], there is evidence of a Th1/Th17 shift and increased IFNγ and IL-17A levels, promoting inflammation and fibrosis [96,97,98]. Inhibiting their integrin-mediated recruitment attenuates liver damage in mice, underscoring their pathogenic role [99].

CD8^+^ T cells play a highly context-dependent role in MASLD. The relative number is elevated in the livers of MASH patients [87], and amplifies liver injury through IFNγ/TNF release and cytotoxicity, although they can support inflammation resolution during regression [100,101,102].

However, MASH-associated mitochondrial dysfunction impairs CD8^+^ T cell mobility and tumor antigen-specific responses [102], facilitating also HCC development.

Finally, although still underexplored, B cells likely contribute to MASLD by producing proinflammatory cytokines (IL-6, TNF) and potentially influencing CD4^+^ T cell polarization toward Th1/Th17 phenotypes [103,104].

Hepatic B cell accumulation and increased anti-oxidation-specific epitopes (OSE) IgG levels have been reported in patients and models [104]. Additionally, a loss of IL-10-producing regulatory B cells has been documented in MASLD models [105].

Interestingly, B cells, particularly IgA^+^ subsets, have been implicated in modulating MDM polarization and promoting fibrogenesis through the transcription factor PU.1 and EF-Hand Domain Family Member D2 (EFHD2)-dependent signaling [106].

Collectively, the above-presented evidence configures a complex scenario where the progression of MASLD is orchestrated by intricate crosstalk between innate and adaptive immune cells, which collectively shape the hepatic inflammatory milieu [4]. On one side, innate immune cells (KCs, neutrophils, and DCs) respond to lipotoxic (and gut-derived) microbial stress by releasing pro-inflammatory cytokines and chemokines that recruit and activate adaptive immune populations [56]. On the other hand, CD4^+^ and CD8^+^ T cells, once primed by antigen-presenting cells, contribute to hepatocellular injury through cytotoxic mechanisms and the secretion of IFNγ and TNF-α [56]. In addition, unconventional T cells—including Mucosal-Associated Invariant T cells (MAIT) and γδ T cells—exhibit dual roles, either exacerbating inflammation or promoting resolution depending on the metabolic and cytokine context [106]. Together, these interactions drive the transition from MASLD to MASH and simultaneously represent promising therapeutic targets for restoring immune homeostasis. Table 2 summarizes the principal implications of immune cells in promoting MASLD to MASH progression (Table 2).

#### 3.2.6. Innate Immune Cell Dysfunction in MASLD-Related Hepatocancerogenesis

The immune system plays a pivotal role in the development of cancer, acting either to suppress tumor growth or, conversely, to support its progression [107]. During the development of HCC, specific molecular pathways trigger unique reactions in particular immune cell subsets, either promoting inflammation or weakening the immune system’s ability to fight the tumor [108].

Hence, various cell types—particularly macrophages and neutrophils—exhibit distinct roles in shaping the tumor microenvironment, reciprocally interacting to create a complex network where HCC progression is promoted [4].

In particular, the HCC microenvironment is characterized by the significant infiltration of macrophages, particularly tumor-associated macrophages (TAMs), predominantly represented by the M2 phenotype [85]. While M1 macrophages are generally antitumoral, they can upregulate PD-L1 expression on HCC cells via IL-1β secretion [85]. In this sense, TAMs promote tumor progression through the secretion of various cytokines (e.g., CSF-1, CCL2) and activation of a signal transducer and activator of transcription 3 (STAT-3), NF-κB, and hypoxia-inducible factor-1 (HIF-1) pathways, secreting immunosuppressive mediators and pro-angiogenic factors that facilitate immune evasion and neovascularization [109,110]. On the other hand, KCs, resident hepatic macrophages, support tumor growth by releasing IL-6, IL-1β, Vascular Endothelial Growth Factor (VEGF), and Platelet-Derived Growth Factor (PDGF), and their activation via TLR signaling contributes to HCC development [85]. Overall, M2 TAMs and KCs are critical in promoting invasion, angiogenesis, and metastasis within the HCC microenvironment [85].

Moreover, the interplay between macrophages and neutrophils within the HCC microenvironment orchestrates a complex immunological landscape that promotes tumor progression [4]. Once considered “minor players” in cancer biology, these cells have recently been shown to exert pro-tumorigenic effects in HCC [108,111,112]. Relevantly, neutrophils, particularly tumor-associated neutrophils (TANs), exhibit functional plasticity and can adopt a pro-tumorigenic N2 phenotype under the influence of TGF-β (and other tumor-derived signals) [108,111,112].

TANs promote angiogenesis at the invasive tumor front via a Matrix Metalloproteinase-9 (MMP-9) release [108,111,112], and chemokines such as CXCL1 and CXCL5 facilitate neutrophil infiltration, correlating with a poor prognosis [109]. In this context, HCC cells stimulate neutrophil-mediated hepatocyte growth factor (HGF) production through GM-CSF, thereby enhancing tumor metastasis via the HGF/c-Met axis [113,114]. Neutrophils also contribute to telomeric DNA damage via ROS production and support tumor development in murine models [85].

Furthermore, NETs have been revealed to promote the transition from MASH to HCC, enhancing inflammation and the metastatic potential [115]. Interestingly, recent evidence has shown that TANs and TAMs engage in reciprocal crosstalk, amplifying inflammatory signaling and remodeling the extracellular matrix to support invasion and metastasis [116]. Moreover, neutrophil-derived ROS and NETs can further polarize macrophages toward a tumor-promoting state, while TAMs modulate neutrophil recruitment and survival via chemokines such as CXCL8 and CCL2 [116]. This bidirectional interaction contributes to the immunosuppressive milieu of HCC and represents a potential target for therapeutic intervention aimed at reprogramming innate immune responses.

Besides macrophages and neutrophils, other immune cells (including DCs, ILCs, and NK/NKT cells) are crucial in MASLD-related cancerogenesis, exerting a distinct and often dual role. DCs play a central role in antigen presentation and antitumor immunity. In HCC, reduced levels of mature CD83^+^ DCs have been observed, simultaneously reporting a positive correlation with a negative prognosis [85]. While IL-12 enhances and IL-10 inhibits the DC function [85], some DC subsets (e.g., intratumoral pDCs and regulatory DCs) have been associated with immunosuppression and poor outcomes via IL-10 and Indoleamine 2,3-dioxygenase (IDO) production [85].

On the other hand, ILCs may exert either pro- or antitumor effects depending on the tumor microenvironment (i.e., levels of certain mediators) and acquisition of certain phenotypes [117]. In HCC, ILC-derived IFNγ has been initially shown to promote hepatocarcinogenesis (in HBV-transgenic mice) [118]. In MASLD-HCC, elevated levels of IL-33 have recently been shown to promote the expansion of ILC2s, which may contribute to tissue repair and anti-inflammatory responses, correlating with improved survival outcomes [119]. Conversely, a shift toward ILC1 dominance under pro-inflammatory conditions may enhance cytotoxicity and simultaneously exacerbate tissue damage and fibrosis [119]. In this sense, the MASLD-HCC microenvironment, often characterized by immunosuppressive signaling and exhausted immune profiles, can reprogram ILCs toward phenotypes that support tumor growth, angiogenesis, and immune evasion [119]. Thus, the functional plasticity of ILCs in MASLD-related HCC underscores their potential as both biomarkers and therapeutic targets in modulating tumor progression.

NK cells and NKT cells also play a pivotal role in tumor immunosurveillance, yet their function is markedly compromised in MASLD-related HCC. In HCC patients, NK cells exhibit phenotypic exhaustion characterized by diminished cytotoxicity and the reduced production of IFNγ, impairing their ability to eliminate malignant hepatocytes and modulate the tumor microenvironment [109,110,120]. This dysfunction is often driven by the upregulation of inhibitory receptors such as CD96 and T cell immunoreceptors with Ig and ITIM domains (TIGIT), and by immunosuppressive cytokines like TGF-β1 within the tumor niche [109,110,120]. Despite these impairments, increased intratumoral NK cell infiltration has been consistently associated with improved prognosis, including longer disease-free and overall survival [109,110,120]. This paradox underscores the importance of the NK cell density and spatial distribution over the mere functional capacity, suggesting that even partially active NK cells may contribute to antitumor immunity via crosstalk with DCs and T cells. Therapeutic strategies aimed at reversing NK cell exhaustion—such as checkpoint blockades or cytokine-based reactivation—hold promise for enhancing innate immune responses and improving clinical outcomes in HCC.

Finally, NKT and, particularly, iNKT cells, a specialized subset of T lymphocytes characterized by the expression of the T Cell Receptor Alpha Variable 10 (TRAV10) gene segment, show a notably reduced concentration in HCC tissues, despite their preserved levels in peripheral circulation [108,109,121]. This intratumoral depletion reflects a localized immunosuppressive milieu and has been associated with poorer clinical outcomes. Indeed, elevated TRAV10 expression within tumor tissue correlates positively with overall and recurrence-free survival, underscoring the prognostic relevance of iNKT cell infiltration and activity [108,109,121].

Mechanistically, tumor-derived transforming growth factor-beta (TGF-β) plays a central role in dampening NKT cell effector functions by downregulating activating receptors and impairing cytokine production, particularly IFNγ [85]. This suppression contributes to immune evasion and tumor progression [108,109,121]. Collectively, this evidence suggests the relevance of restoring the iNKT cell function or mitigating TGF-β-mediated inhibition as promising strategies for enhancing antitumor immunity in HCC.

Table 3 briefly summarizes the principal implications of immune cells contributing to MASLD-related cancerogenesis (Table 3).

## 4. “Trained” Immunity in the Progression of MASLD

### 4.1. Immunometabolism and Trained Immunity: New Insights in Immune Regulation

In recent years, a new area of research termed “immunometabolism” has emerged, highlighting the intricate connection between immunology and metabolism [122,123,124,125]. The interplay between these two systems encompasses multiple dimensions. Immune cells residing in adipose tissue and the liver regulate essential metabolic functions, such as lipolysis and insulin action [122,123,124,125]. Moreover, the nutrient intake and metabolism—particularly of sugars, fats, and proteins—can modulate immune responses [123]. In this context, chronic metabolic inflammation, also known as “metaflammation,” has been identified as a central hallmark of metabolic disorders [126]. This inflammation, driven by cellular and molecular components of the immune system, involving adipose tissue, the liver, and the pancreas, contributes to the onset and progression of obesity-associated conditions, including MASLD [122,126,127]. Notably, the accumulation and activation of macrophages in these tissues during obesity play a critical role in metabolic dysfunction [122,124]. Their function is shaped by metabolic adaptations to the lipid-rich environment [95].

In support of this, single-cell analyses have recently identified a macrophage subpopulation with a lipid metabolism-associated signature, termed “lipid-associated macrophages,” enriched in obese adipose tissue and the liver [128,129].

An additional key aspect of immunometabolism is that cellular metabolism directly regulates immune cell activation and function [122,130]. In particular, distinct metabolic states underlie different activation states of innate immune cells, supporting their phenotype and functional plasticity [122,130].

The metabolic reprogramming of innate immune cells represents the essential epiphenomenon of innate immune memory, constituting a cornerstone of TI theory [26]. According to this, immunological memory, once thought to be an exclusive feature of the adaptive immune system, also pertains to the innate immune arm [30]. Innate immune cells have been shown to respond to subsequent encounters with unrelated antigenic stimuli—whether exogenous [25], endogenous, or metabolism-derived—by activating distinct signaling pathways, determining the epigenetic remodeling and the reprogramming of multiple intracellular metabolic routes [24,30].

TI thus provides a form of memory within the innate immune system, resulting in the long-term functional reprogramming of innate immune cells [104]. Within this complex framework, immune cell activation induces gene expression reprogramming, resulting in the acquisition of novel functional capabilities such as cytokine secretion, lipid mediator production, a tissue remodeling enzyme release, migratory capacity, and cell proliferation [29].

There is growing interest in the metabolic control of immune cell activation, as well as its implications in tumorigenesis. Understanding the bidirectional relationship between metabolism and immunity could shed light on the pathophysiology of metabolic and inflammatory diseases, as well as open new avenues for therapeutic interventions targeting immune cell metabolism [30].

### 4.2. Immunometabolic Pathways Contributing to MASLD/MASH Progression

The MASLD/MASH pathogenesis is intricately linked to the interplay between metabolic disturbances and immune system alterations, particularly within the hepatic microenvironment [30].

In MASLD/MASH scenarios, an excessive nutrient intake, synergistically with IR, represents the primum movens contributing to hepatic FFAs’ deposition [131]. As previously shown, increased lipogenesis and reduced FFAs beta-oxidation lead to fat accumulation in hepatocytes, resulting in the formation of toxic lipid intermediates such as saturated fatty acids, ceramides, and free cholesterol [132]. These intermediates, acting as DAMPs, can activate innate immune cells, promoting, as a first consequence, inflammation via inducing NF-κB and the NLRP3 inflammasome-related pathways [133]. In particular, these lipotoxic signals are sensed by innate immune cells, particularly hepatic macrophages, which respond by releasing pro-inflammatory cytokines and chemokines that perpetuate tissue injury and recruit adaptive immune effectors [133].

On the other hand, the accumulation of hepatotoxic lipids—such as ceramides, diacylglycerols, and free cholesterol—triggers endoplasmic reticulum (ER) stress, mitochondrial dysfunction, and oxidative damage, leading to hepatocyte damage, inflammation, and apoptosis [31]. This condition, in turn, impairs the mitochondrial function, causing the release of mitochondrial DNA and other DAMPs, which—via NLRP3 inflammasome—promote the secretion of pro-inflammatory cytokines IL-1β and IL-18 [131,134]. Concurrently, mitochondrial dysfunction leads to increased ROS production, which further exacerbates OS and cellular damage [131,134]. From this landscape, the lipotoxic metabolites, DAMPs, and elevated ROS levels emerge as immunogenic signals, inducing the innate immune pathways and thus inflammation.

These mechanisms constitute the “traditional” immunity contribution to the MASLD progression. However, these processes promoting and sustaining phlogosis, in the absence of dramatic metabolic shifts impacting energetic availability, appear to be potentially destined to run out.

Revolutionarily, in recent decades, chronic exposure to these same stimuli has also been shown to represent a crucial trigger-enhancing TI response with relatively immunometabolic and reprogramming consequences [27]. In this sense, lipotoxic intermediates, DAMPs, and elevated ROS levels have all been implicated in the induction of TI, contributing to sustaining an enhanced (i.e., impaired) inflammatory response upon secondary unspecific stimulation [27]. This phenomenon determines metabolic reprogramming (including, among others, alterations in glycolysis) [135], which collectively contributes to perpetuating the inflammatory status in MASH via a changing cellular energetic profile [32].

Within this context, macrophages emerge as key players, as their polarization and function are tightly regulated by the same metabolic immunogenic triggers that drive TI [136]. In the inflammatory MASH milieu, macrophages predominantly adopt a pro-inflammatory (M1) phenotype characterized by increased glycolysis and oxidative phosphorylation (OXPHOS) [132,136]. This metabolic shift supports the rapid ATP generation and biosynthesis necessary for inflammatory responses [137,138]. Conversely, anti-inflammatory (M2) macrophages’ metabolic activities primarily contribute to FFAs’ oxidation [139]. The balance between these phenotypes depends on bioenergetic cell availability and is regulated by metabolic pathways, including mitochondrial oxidative phosphorylation, and, most of all, the tricarboxylic acid (TCA) cycle (i.e., Krebs cycle) [140].

In support of this, the inhibition of oxidative phosphorylation has initially been shown to prevent the development of a trained macrophage phenotype, highlighting the importance of the mitochondrial function in the macrophage polarization and function [141]. Subsequently, considering the centrality of energy metabolism in immune regulation, accumulating research has focused on the bioenergetic remodeling of immune cells in MASLD. Recent findings have identified a distinct metabolic signature in circulating monocytes from both murine MASH models and patients with MASH, characterized by the concurrent upregulation of mitochondrial oxidative phosphorylation and glycolytic flux [142]. This dual enhancement of energy pathways in macrophages disrupts the redox balance, leading to the excessive production of ROS and pro-inflammatory cytokines, thereby accelerating hepatic injury and disease progression [142].

Among the transcriptional regulators implicated in this process, E2F Transcription Factor 2 (E2F2), a member of the E2F family known for its role in cell cycle control and apoptosis, has emerged as a key modulator of macrophage immunometabolism [143]. Notably, E2F2 expression is markedly reduced in hepatic macrophages during MASH, and its deficiency has been shown to exacerbate liver inflammation, HSC activation, and lipid accumulation [144]. Mechanistic investigations revealed that a loss of E2F2 promotes a pro-inflammatory macrophage phenotype via a leucine–mTORC1-dependent pathway, wherein the increased expression of the amino acid transporter Solute Carrier Family 7 Member 5 (SLC7A5) enhances glycolysis and impairs the mitochondrial function [144]. These findings underscore the importance of E2F2 in maintaining metabolic homeostasis and suggest that restoring its activity may represent a promising therapeutic strategy for MASLD.

In addition to this factor, potentiating the expression of activating transcription factor 3 (ATF3) has been reported to improve glucolipid metabolism in liver macrophages, ultimately protecting against MASH development [145]. ATF3, a stress-induced transcription factor binding to the cyclic AMP response element (CRE) [146], represents a key regulator of the glucose—fatty acid cycle and promotes FFAs oxidation in macrophages via a reduction in cellular glucose levels by inhibiting head box 1 (FoxO1)-mediated gluconeogenesis, as well as FFA uptake by CD36 [146].

Relevantly, AFT3 expression levels were down-regulated in liver macrophages and negatively correlated with MASH severity, suggesting that correcting this aberrant metabolic pathway in macrophages may be a plausible approach for MASLD [145].

Moreover, β-arrestin 2, a multifunctional adaptor protein for the desensitization and internalization of G protein-coupled receptors (GPCRs) [147], was found to be elevated in liver macrophages and circulating monocytes in patients with MASH. Interestingly, β-arrestin 2 in myeloid cells promoted the ubiquitination of immune responsive gene 1 *(IRG1*), leading to decreased itaconate production and increased succinate dehydrogenase activity in macrophages. These dysregulated Krebs cycle metabolites fuel the release of mitochondrial ROS and M1 polarization, ultimately exacerbating MASH. Therefore, also targeting β-arrestin 2 may be a potential strategy for MASH treatment [142].

Collectively, the above-presented metabolic rewiring reinforces the inflammatory hepatic environment, contributing to the chronic inflammation and disease progression in MASH [148]. However, in addition to the above-presented repercussions, the induction of the aberrant TI response also determines epigenetic changes that enhance the inflammatory potential of innate immune cells [136].

In MASLD/MASH dysmetabolic/dyslipidaemic oxidative stress scenarios, chronic exposure to oxidized low-density lipoprotein (oxLDL) profoundly alters the functional and metabolic landscape of monocytes and macrophages [32,141]. OxLDL acts as DAMPs, engaging pattern recognition receptors such as TLR2 and TLR4, and initiating signaling cascades that upregulate glycolytic enzymes [including GLUT1, hexokinase 2 (HK2), and 6-phosphofructo-2-kinase/fructose-2,6-biphosphatase 3 (PFKFB3)] [32,141]. This glycolytic shift is accompanied by increased oxygen consumption and mitochondrial remodeling, with oxLDL-trained macrophages exhibiting enlarged mitochondria and elevated OXPHOS activity [32,141]. The accumulation of TCA cycle intermediates—particularly succinate and fumarate—further amplifies inflammation by stabilizing HIF-1α and promoting ROS production [32,141]. Importantly, these metabolites also serve as epigenetic modifiers, inhibiting histone demethylases and leading to histone hypermethylation at promoters of the pro-inflammatory genes IL-6 and TNF-α [32,141].

This epigenetic remodeling sustains cytokine hyper-responsiveness and reinforces the TI phenotype. In support of this, the pharmacological inhibition of OXPHOS has been shown to abrogate oxLDL-induced TI, highlighting the relative therapeutic potential of targeting this pathway [149].

Interestingly, emerging findings have demonstrated that the pharmacological modulation of intracellular glutathione levels directly influences the magnitude of pro-inflammatory cytokine production, particularly IL-1β and TNF-α, in human monocytes exposed to trained immunity stimuli such as β-glucan and oxLDL [150]. Mechanistically, glutathione synthesis supports the metabolic and epigenetic reprogramming required for the establishment of trained immunity, including enhanced glycolysis, glutaminolysis, and histone modifications at immunometabolic gene loci. The inhibition of glutamate–cysteine ligase (GCLC), the rate-limiting enzyme in glutathione biosynthesis, impairs H3K27me3 demethylation and reduces the accessibility of promoters for key inflammatory genes, thereby attenuating the trained phenotype [141]. Moreover, glutathione levels have been positively correlated with IL-1β production in BCG-vaccinated individuals, suggesting a functional link between the antioxidant capacity and innate immune memory [141]. These findings underscore the importance of glutathione as a metabolic integrator of immune training and highlight its potential as a therapeutic target in hyperinflammatory conditions such as MASLD and MASH.

Finally, recent evidence has also implicated Liver X Receptor alpha (LXRα) as a key regulator of oxLDL-induced TI [151]. The activation of LXRα enhances inflammatory cytokine production and histone acetylation at inflammatory gene loci, whereas its inhibition reverses metabolic and epigenetic reprogramming [151].

These results underscore the role of nuclear receptor signaling in modulating innate immune memory and suggest that LXRα may be a viable target for dampening chronic inflammation in MASH.

Altogether, the above-presented findings reveal that DAMPs-induced metabolic and epigenetic reprogramming in macrophages contributes to the persistence of hepatic inflammation and fibrosis in MASH. Understanding the molecular underpinnings of TI in this context opens new avenues for therapeutic intervention aimed at restoring immune tolerance and metabolic balance.

Figure 1 summarizes the main immunometabolic pathways contributing to MASLD/MASH progression (Figure 1).

### 4.3. Immunometabolic Pathways Driving Hepatocellular Carcinoma Progression

Immunometabolic reprogramming constitutes a fundamental feature of cancer biology, empowering malignant cells to satisfy the augmented bioenergetic and biosynthetic demands that underpin uncontrolled proliferation, tissue invasion, and metastatic dissemination [85]. As the principal metabolic organ, the liver integrates and coordinates core pathways involved in carbohydrate, lipid, and amino acid metabolism [152]. Therefore, in the MASLD context, hepatocancer onset represents the tip of the iceberg of a complex pathogenetic scenario where HCC develops and progresses in a heterogeneous microenvironment shaped by chronic inflammation, OS, and profound immunometabolic dysregulation [153]. In this landscape, immune cells play a pivotal role, not merely responding to tumor cells, but actively sculpting the conditions under which they thrive [154]. Recent evidence underscores how the metabolic phenotype of the hepatic immune cells directly governs their function and fate, influencing both tumor surveillance and tumor promotion [153,154]. In this sense, metabolic stress within the tumor microenvironment exerts profound immunomodulatory effects, impairing the cytotoxic function of effector immune cells and facilitating immune evasion in HCC [152]. As already described above, macrophage polarization is tightly linked to intracellular metabolic pathways. M1 macrophages—typically pro-inflammatory—display high glycolytic flux, enhanced fatty acid synthesis, and the increased activity of the pentose phosphate pathway (PPP), while downregulating oxidative phosphorylation (OXPHOS) and TCA cycle activity. This metabolic reprogramming supports the production of ROS, pro-inflammatory cytokines, and antimicrobial functions [154,155]. In contrast, M2 macrophages adopt an anti-inflammatory phenotype sustained by OXPHOS and fatty acid oxidation (FAO) with reduced glycolysis, facilitating tissue remodeling and immunosuppression [154]. Interestingly, TAMs shift dynamically between these phenotypes, adapting to the metabolic demands of the tumor microenvironment [156]. In early HCC, TAMs rely primarily on glycolysis, whereas at advanced stages, they transition to OXPHOS-dominant metabolism [157]. A central mediator of this dynamic shift is lactate, the principal product of aerobic glycolysis, which accumulates due to the Warburg effect [158,159].

The lactate, produced also by tumor cells, activates HIF-1α, enhancing the expression of pro-tumorigenic genes [including VEGFA and Arginase 1 (ARG1)] [160]. Notably, increased glycolysis in TAMs correlates with elevated PD-L1 expression via the upregulation of 6-Phosphofructo-2-Kinase/Fructose-2,6-Biphosphatase 3 (PFKFB3), contributing to immune evasion [161]. These changes not only favor angiogenesis and extracellular matrix remodelling, but also suppress cytotoxic immune responses. Moreover, elevated lactate concentrations have been shown to upregulate IL-23 and IL-12, activating the IL-23/IL-17 axis and fostering a pro-inflammatory yet immunosuppressive milieu [162]. In addition, the lactate modulates the activity of the nuclear factor of the activated T-cells (NFAT) in both T and NK cells, leading to reduced IFNγ production and diminished antitumor immunity [163]. Clinically, serum lactate levels correlate with the tumor burden in HCC patients, underscoring its role as both a metabolic and immunological biomarker [164].

In parallel, nutrient competition within the tumor microenvironment—particularly for glucose and glutamine—emerges as a critical determinant of immune competence. T cells rely heavily on these substrates to sustain their effector functions, yet the heightened metabolic demands of tumor cells deprive immune cells of essential nutrients, thereby blunting their antitumor capacity [164]. Glycolytic reprogramming in cancer cells further contributes to immune escape by inducing Fas ligand (FasL) expression, which triggers activation-induced cell death (AICD) in T cells via TCR restimulation [163]. Intriguingly, a positive correlation has been observed between glucose metabolic reprogramming and elevated alpha-fetoprotein (AlphaFP) expression in liver tumors [163]. Given AFP’s known ability to inhibit Fas expression and suppress immune cell function, it has been hypothesized that glucose-driven AFP upregulation may represent a strategic adaptation by hepatic cancer cells to evade immune surveillance [165].

Figure 2 summarizes the main glucose metabolism-related immunometabolic pathways contributing to MASLD-related HCC progression (Figure 2).

Beyond glucose metabolism, lipid pathways are equally influential in mediating metabolic-guided cancerogenesis [152].

Relevantly, beyond hepatocytes, LDs are also abundant in immune cells like KCs and infiltrating macrophages, where they serve as platforms for the synthesis of bioactive lipid mediators, including prostaglandins and leukotrienes [31,86]. These lipid-derived signals further amplify the inflammatory response and contribute to immune cell recruitment and activation. Moreover, LDs support the metabolic reprogramming of immune cells, fostering a pro-inflammatory phenotype associated with TI [31,86]. Regarding this, arachidonic acid metabolism and the overproduction of prostaglandin E2 (PGE2) in HCC have been shown to suppress antitumor immunity via the Prostaglandin E 2 receptor 4 (EP4) receptor on immune cells, inhibiting IFNγ and TNF-α while promoting IL-10 and IL-6 production [166]. In contrast, the role of cholesterol remains controversial, considering how high serum levels may enhance NK cell activity and restrict tumor growth, simultaneously impairing cytotoxic T cell function [167].

Amino acid metabolism also shapes the immune milieu in HCC. Elevated asparagine metabolism is associated with an immunosuppressive tumor microenvironment, marked by increased Alanine/Serine/Cysteine Transporter 2 (ASCT2) and decreased Glutaminase 2 (GLS2) expression [168,169]. This shift promotes the infiltration of regulatory T cells and M2 macrophages while reducing M1 macrophages and effector T cell populations, correlating with poor prognosis [170].

The above-presented metabolic adaptations are deeply influenced and sustained by OS, a shared hallmark of chronic liver disease. In the MASH scenario, the overloading of FFAs β-oxidation leads to electron leakage and ROS generation [171,172]. ROS, while initially signaling molecules, can cause DNA damage, lipid peroxidation, and the sustained activation of NF-κB and AP-1, thereby promoting carcinogenesis [173].

Moreover, these molecules, by acting as DAMPs, represent potent activators of the NLRP3 inflammasome, particularly within KCs [173]. Once activated, NLRP3 promotes the caspase-1-mediated cleavage of pro-IL-1β and pro-IL-18 into their active forms—cytokines that stimulate hepatocyte proliferation, fibrogenesis, and immune cell recruitment [174]. Once again, this process is not merely transient and may persist due to an aberrant TI response and relative metabolic changes. In particular, HIF-1α—stabilized by both hypoxia and ROS—further drives glycolytic gene expression, thereby promoting tumor aggressiveness [175,176].

Collectively, these mechanisms configure a vicious circle where MASLD-related (lipo)toxic stimuli and DAMPs trigger aberrant TI activation, determined as collateral consequences, exalted ROS production, which appears able in turn to sustain innate immune dysfunction, ultimately creating a microenvironment where the reciprocal perpetuation of inflammation and OS promotes HCC onset and progression.

## 5. Future Perspectives: Modulating Immunometabolism as a Promising Strategy in the Management of MASLD/MASH

Recent advances have expanded the classical landscape of immunopathogenesis in MASLD by introducing the novel concept of TI, revolutionarily supporting a form of memory in innate immune cells driven by epigenetic and metabolic reprogramming after the first stimulus [28]. In the hepatic setting, canonical trainers such as LPS, β-glucan, oxidized LDL, or excess FFAs can “prime” KCs and infiltrating monocytes to mount an exaggerated, non-specific response upon secondary challenges. Evidence from in vitro models and patient samples indicates that TI skews hepatic macrophages toward a hyper-responsive IL-1β/TNF-α phenotype and amplifies ROS generation—mechanistic hallmarks that fit precisely with the chronic, low-grade inflammation of MASLD [27].

Moreover, metabolite-induced TI integrates seamlessly with the emerging field of immunometabolism: rewired glycolysis and the mevalonate pathway supply acetyl-CoA for histone acetylation, locking pro-inflammatory genes in an open chromatin state and perpetuating chemokine production long after the original trigger has waned. This persistent inflammatory “scar” provides a plausible explanation for why metabolic hits (overnutrition, dysbiosis, and IR) continue to drive steatohepatitis and fibrogenesis even in the absence of classical pathogens, thereby bridging metabolic dysfunction with progressive liver injury and, ultimately, oncogenesis. In short, the liver’s position at the crossroads of metabolism and immunity—together with the capacity of innate cells to acquire TI features—places immunometabolic rewiring at the forefront of MASLD pathogenesis [27]. Emerging evidence underscores the central role of immunometabolism in the progression of MASLD, positioning it as a novel and promising therapeutic frontier.

Among the most compelling developments is the pharmacological inhibition of ATP citrate lyase (ACLY), a pivotal enzyme linking glycolysis and de novo lipogenesis. The recent clinical approval of bempedoic acid for the treatment of hypercholesterolemia marks a significant therapeutic advance, offering anti-inflammatory benefits through reductions in high-sensitivity C-reactive protein (hs-CRP) levels, independently of lipid-lowering effects [177,178]. Despite this progress, several ACLY inhibitors remain limited by suboptimal pharmacokinetics, poor selectivity, or off-target toxicity, emphasizing the need for next-generation compounds with improved efficacy and safety profiles [179]. In parallel, targeting isocitrate dehydrogenase 1 (IDH1) mutations has shown therapeutic relevance in intrahepatic malignancies, including cholangiocarcinoma (iCCA). The approval of ivosidenib, a selective IDH1 inhibitor, for chemotherapy-refractory IDH1-mutant iCCA represents a major milestone in precision oncology [180,181].

Additional IDH1 inhibitors (i.e., olutasidenib, LY3410738, and HMPL-306) are currently under clinical evaluation [182]. Moreover, IDH1-mutant tumors are being explored as candidates for synthetic lethality strategies involving poly (ADP-ribose) polymerase (PARP) inhibitors, exploiting underlying DNA repair deficiencies [183].

In this sense, leveraging the principles of TI represents a promising immunotherapeutic strategy in hepatic cancer, particularly in advanced, unresectable HCC. By reprogramming innate immune cells—such as monocytes, macrophages, and DCs—TI-based interventions aim to overcome tumor-induced immune resistance and restore effective antitumor responses [184]. Notably, BCG vaccination, traditionally employed against tuberculosis, has demonstrated superior antitumor efficacy compared to a PD-1 blockade in preclinical HCC models [185]. This enhanced activity is attributed to the robust recruitment of M1-polarized macrophages and effector T cells into the tumor microenvironment, accompanied by elevated IFNγ production and reduced T cell exhaustion [185].

In parallel, a recent phase I/II clinical trial evaluating a personalized DNA neoantigen vaccine (GNOS-PV02) in combination with pembrolizumab reported encouraging results in advanced HCC patients [186]. Approximately 30% of participants exhibited objective tumor shrinkage, including three complete responses, with no dose-limiting toxicities. Mechanistically, this approach induces neoantigen-specific CD4^+^/CD8^+^ T cell responses, expands cytotoxic T cell clones, and promotes their trafficking into the tumor [186]. Together, these findings underscore the therapeutic potential of TI-based modalities to reshape the immunological landscape of hepatic cancer [187]. By integrating innate immune reprogramming with neoantigen-directed vaccination and checkpoint inhibition, it may be possible to elicit durable and systemic antitumor immunity in otherwise refractory disease settings [187].

Looking forward, the integration of immunometabolic biomarkers into diagnostic algorithms will enable a shift toward precision medicine, allowing for stratification and individualized treatment approaches based on specific metabolic and genetic profiles [188]. Furthermore, the early identification of immunometabolic dysregulation may pave the way for preventive medicine strategies aimed at halting disease progression in at-risk individuals before the onset of irreversible hepatic or oncogenic complications [188].

## 6. Conclusions

The modern paradigm of TI provides novel insights into the chronic activation of innate immune cells, while immunometabolic remodeling underscores the significance of bioenergetic control in shaping immune cell functionality, creating a scenario where the dynamic interplay between lipotoxic stress, oxidative damage, and epigenetic reprogramming orchestrates a sustained inflammatory state that drives hepatic fibrogenesis and carcinogenesis, leading to MASH, fibrosis, and HCC.

The continued exploration of these mechanisms may pave the way for innovative therapeutic strategies aimed at restoring immune homeostasis and interrupting disease MASLD progression.

## Figures and Tables

**Figure 1 biomedicines-13-02004-f001:**
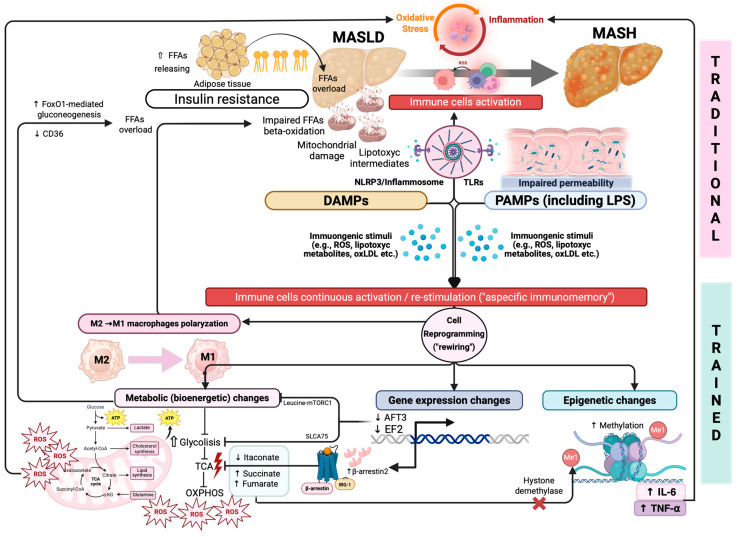
Main “Traditional” and “Trained” immunity pathways sustaining MASLD to MASH progression. In early stages, hepatocyte lipid accumulation and oxidative stress activate traditional innate immune pathways, promoting cytokine release and inflammasome activation. Persistent metabolic and microbial stimuli induce trained immunity, characterized by epigenetic and metabolic reprogramming of innate immune cells. These reprogrammed cells exhibit heightened inflammatory and fibrogenic responses, perpetuating hepatic injury and driving progression from steatosis to steatohepatitis. MASLD: metabolic dysfunction-associated steatotic liver disease; MASH: metabolic dysfunction-associated steatohepatitis; TCA: tricarboxylic acid; TLR: toll-like receptors; FFAs: free fatty acids; PAMPs: pathogen-associated molecular patterns; DAMPs: damage-associated molecular patterns; ROS: reactive oxygen species; OXPHOS: oxidative phosphorylation; SLCA75: Solute Carrier Family 7 Member 5; EF2: Transcription Factor 2; OxLDL: oxidized low-density lipoprotein; AFT3: activating transcription factor 3; IRG-1: immune responsive gene 1; Fox-O1: head box 1; IL: interleukin; TNF: tumor necrosis factor.

**Figure 2 biomedicines-13-02004-f002:**
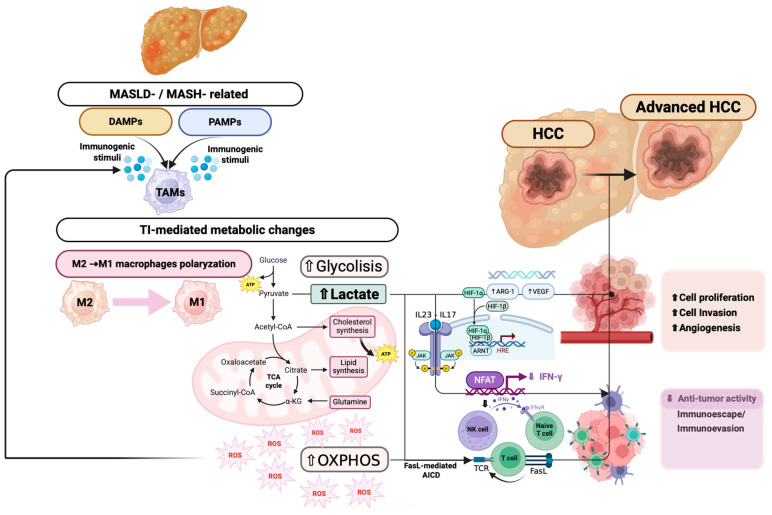
Key glucose-metabolism related immunometabolic pathways promoting hepatocancerogenesis in MASLD. MASLD: metabolic dysfunction-associated steatotic liver disease; MASH: metabolic dysfunction-associated steatohepatitis; TCA: tricarboxylic acid; PAMPs: pathogen-associated molecular patterns; DAMPs: damage-associated molecular patterns; ROS: reactive oxygen species; OXPHOS: oxidative phosphorylation; TCR: T-cell receptor; IFN: interferon; AICD: activation-induced cell death; NFAT: nuclear factor of activated T-cells; HIF: hypoxia-inducible factor; HRE: hormone response element; ARNT: aryl hydrocarbon receptor nuclear translocator; ARG-1: Arginase 1; VEGF: vascular endothelial growth factor; NK: natural killer cells; IL: interleukin; HCC: hepatocellular carcinoma.

**Table 1 biomedicines-13-02004-t001:** “Traditional Immunity” response vs. “Trained immunity” response in MASLD progression.

Feature	Traditional Immunity	Trained Immunity	References
**Definition**	Immediate, non-specific immune response to pathogens or damage	Long-lasting functional reprogramming of innate immune cells after initial stimulus	[4,24,26]
**Memory Formation**	No immunological memory	Epigenetic and metabolic memory-like features	[26,29]
**Response Specificity**	Non-specific, same response to repeated stimuli	Enhanced response upon re-exposure to similar or unrelated stimuli	[30]
**Duration of Effect**	Short-lived	Persistent (weeks to months)	[4,30,31]
**Mechanism of Activation**	Pattern recognition receptors (PRRs) detecting PAMPs/DAMPs	PRRs plus metabolic and epigenetic reprogramming	[4,27]
**Role in MASLD**	Initial inflammation, cytokine release, immune cell recruitment	Sustained inflammation, fibrosis, progression to MASH and HCC	[4,27]
**Therapeutic Implications**	Targeting acute inflammation	Modulating trained immunity to prevent chronic liver damage	[14,32]

DAMPS: damage-associated molecular patterns; PAMPS: pathogen-associated molecular patterns; MASH: metabolic dysfunction-associated steatohepatitis; HCC: hepatocellular carcinoma.

**Table 2 biomedicines-13-02004-t002:** Pathogenetic contribution of immune cells in MASLD to MASH progression.

Cell Type	Principal Implications in MASLD/MASH Pathogenesis	References
**Macrophages**	Promote hepatic inflammation (via cytokines, ROS) and fibrosis; recruitment of monocyte-derived macrophages (MDMs) exacerbates liver injury. YAP and miR-204-3p represent the main implicated pathways.	[37,40,41,42]
**Neutrophils**	Dual role: exacerbate inflammation via NE, MPO, and NETs; contribute to insulin resistance, steatosis, and fibrosis; miR-223 mediates protective effects.	[12,43,44,45,46,47,48,49,50,51]
**Dendritic cells (DCs)**	Shift toward proinflammatory phenotype with lipid overload; c-kit^+^ cDC1 cells exert protective effects; LKB1-AMPK/SIK pathway restrains Th17 cells.	[30,37,52,53,54]
**Natural killer (NK) cells**	Increase in NK activation during MASLD; phenotypic shift toward ILC1-like cells influences inflammation and disease progression.	[37,55,56,57,58,59,60]
**Natural killer T (NKT) cells**	Dual role: exacerbate inflammation and fibrosis in steatohepatitis but may protect against fibrosis in certain models.	[61,62,63,64,65]
**CD4^+^ T cells**	Polarization toward Th1/Th17 phenotypes drives MASLD progression; IFNγ and IL-17 production promote inflammation and fibrosis.	[69,70,71,72,73,74]
**CD8^+^ T cells**	Amplify liver inflammation via IFNγ and TNF; cytotoxic activity drives hepatocellular damage; aid in resolution during regression phases.	[62,75,76,77]
**B cells**	Promote inflammation through cytokines (IL-6, TNF); elevated anti-OSE IgG; regulatory B cell loss exacerbates disease.	[78,79,80]

ROS: reactive oxygen species; NE: neutrophil elastase; MPO: myeloperoxidase; NETs: neutrophil extracellular traps; YAP: yes-associated protein; LKB1-AMPK/SIK: liver kinase B1-activated protein kinase/salt-inducible kinase; IL: interleukin; IFN: interferon; TNF: Tumor Necrosis Factor; OSE: oxidation-specific epitopes.

**Table 3 biomedicines-13-02004-t003:** Innate and innate-like immune cells in the pathogenesis of hepatocellular carcinoma (HCC).

Cell Type	Principal Involvement in HCC Pathogenesis	References
**Macrophages (TAMs)**	M2 TAMs promote tumor growth via cytokines (CSF-1, VEGF, CCL2) and STAT-3, NF-κB, and HIF-1; M1 macrophages may upregulate PD-L1; Kupffer cells (KCs) release IL-6, IL-1β, VEGF, and PDGF and support tumor progression via TLR signaling.	[85,92]
**Neutrophils (TANs)**	Promote angiogenesis via MMP-9; infiltrate via CXCL1/CXCL5; produce HGF (stimulated by GM-CSF), enhancing metastasis via HGF/c-Met axis; induce DNA damage via ROS; form NETs that support inflammation and metastasis.	[82,83,84,85,86,87,88]
**Dendritic Cells (DCs)**	Reduced mature CD83^+^ DCs linked to prognosis; IL-12 enhances and IL-10 inhibits DC function; pDCs and regulatory DCs promote immunosuppression via IL-10 and IDO.	[85]
**Innate-like T Cells (ILCs)**	ILC-derived IFNγ promotes hepatocarcinogenesis; NK cells show reduced function but correlate with good prognosis when infiltrating; NKT cells reduced in HCC tissue; TGF-β suppresses NK/NKT activity.	[82,85,89,90,91,92,93]
**MAIT Cells**	Reduced in tumor core; high intratumoral density linked to poor prognosis; may gain protumor functions during HCC development.	[82,94]
**γδ T Cells**	Involved in early surveillance; low numbers linked to recurrence; may promote tumor growth via IL-17.	[82]

TAMs: tumor-associated macrophages; TANs: tumor-associated neutrophils; MAIT: Mucosal-Associated Invariant T cells; IL: interleukin; CSF-1: colony-stimulating factor 1; VEGF: Vascular Endothelial Growth Factor; CCL2: chemokine (C-C motif) ligand 2; PDGF: Platelet-derived Growth Factor; TLR: toll-like receptor; MMP-9: Matrix Metalloproteinase-9; NETs: neutrophil extracellular traps; IDO: Indoleamine 2,3-dioxygenase; IFN: interferon; ROS: reactive oxygen species; GM-CSF: Granulocyte–Macrophage Colony-Stimulating Factor; HGF: Hepatocyte Growth Factor.

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
