# Peer review of "From “Traditional” to “Trained” Immunity: Exploring the Novel Frontiers of Immunopathogenesis in the Progression of Metabolic Dysfunction-Associated Steatotic Liver Disease (MASLD)"

_biomedicines, 2025, doi:10.3390/biomedicines13082004_

Round 1
Reviewer 1 Report
Comments and Suggestions for Authors
In the current manuscript in the journal Biomedicines (Manuscript ID: Biomedicines-3807245) with a special issue on Oxidative Stress and Inflammation in Non-communicable Diseases, Mario Romeo et al. have reviewed on the impact of the most prevalent chronic liver disease called as Metabolic dysfunction-associated steatotic liver disease (MASLD) on the hepatocellular carcinoma and other associated liver pathogenesis. This article compiles that in MASLD how the insulin resistance driven metabolic imbalance accredited to a multifactorial pathogenesis involving chronic inflammation, oxidative stress and immune dysregulation.
Overall, this is a well compiled article that offers significant insights into the current understanding of how oxidative damage, lipotoxic stress, and epigenetic reprogramming interact dynamically to create a persistent inflammatory state that promotes hepatic fibrogenesis and cancer.
The article is well written; English is good and includes all the important information. The tables cover the significant details of the immune checkpoints, and the figures explain the clear immunometabolism pathways promoting hepatocarcinogenesis.
Author Response
In the current manuscript in the journal Biomedicines (Manuscript ID: Biomedicines-3807245) with a special issue on Oxidative Stress and Inflammation in Non-communicable Diseases, Mario Romeo et al. have reviewed on the impact of the most prevalent chronic liver disease called as Metabolic dysfunction-associated steatotic Liver Disease (MASLD), on the hepatocellular carcinoma and other associated liver pathogenesis. This article compiles that in MASLD how the insulin resistance-driven metabolic imbalance accredited to a multifactorial pathogenesis involving chronic inflammation, oxidative stress, and immune dysregulation.
Overall, this is a well-compiled article that offers significant insights into the current understanding of how oxidative damage, lipotoxic stress, and epigenetic reprogramming interact dynamically to create a persistent inflammatory state that promotes hepatic fibrogenesis and cancer.
The article is well written; English is good and includes all the important information. The tables cover the significant details of the immune checkpoints, and the figures explain the clear immunometabolism pathways promoting hepatocarcinogenesis.
Reply: We sincerely thank the Reviewer for His/Her evaluable comments and we are very glad that He/She has appreciated our manuscript.

Reviewer 2 Report
Comments and Suggestions for Authors
The review is composed well and touched the many aspects of the development of MASLD. However, a few aspects need to be incorporated in this manuscript.
- What is traditional and trained immunity. Authors should add few line in the abstract section to provide more meaning of it.
- Is that the trained immunity good for preventing from the MALSD progression?
- Highlight the key point of the MALSD progression in the conclusion.
- How lipid droplets are involved in the MASLD progression.
- Write few sentences of the legends of the Figure 1. To reflect that what is happening in traditional and trained immunity.
Author Response
Reviewer 2
The review is composed well and touched the many aspects of the development of MASLD.
However, a few aspects need to be incorporated in this manuscript.
- What is traditional and trained immunity? Authors should add few line in the abstract section to provide more meaning to it.
Reply: We sincerely thank the Reviewer for this precious comment. As suggested, in the resubmitted version of our manuscript, a few lines clarifying what is traditional immune response is and what trained immunity is have been added.
2.Is that the trained immunity good for preventing from the MALSD progression?
Reply: We thank the Reviewer for asking to clarify this cornerstone point.
Trained immunity (TI), while initially perceived as a beneficial enhancement of innate immune responses, appears to play a dual role in the context of MASLD. On one hand, TI can bolster host defense mechanisms against pathogens and promote tissue repair. However, in MASLD, this heightened immune responsiveness often becomes maladaptive. Persistent metabolic and microbial stimuli—such as excess free fatty acids and gut-derived endotoxins—can induce epigenetic and metabolic reprogramming in innate immune cells like Kupffer cells and monocytes. These "trained" cells then exhibit exaggerated inflammatory responses, producing pro-inflammatory cytokines, lipid mediators, and fibrogenic enzymes that amplify hepatic inflammation and fibrosis.
Thus, rather than preventing MASLD progression, trained immunity may accelerate the transition from simple steatosis to steatohepatitis, fibrosis, and even hepatocellular carcinoma. Targeting the maladaptive aspects of TI and its immunometabolic drivers could offer promising therapeutic strategies to halt or reverse disease progression.
In the resubmitted version of our manuscript, all these features and concepts have been properly discussed in the introduction section dedicated to the presentation of (TI) in MASLD. Moreover, the potential applications deriving from modulating specific TI-targets and the relative therapeutic implications have been properly discussed in the paragraph dedicated to future perspectives.
3. Highlight the key point of the MALSD progression in the conclusion.
Reply: We thank the Reviewer for this suggestion. In the resubmitted version of our manuscript, the conclusion section has been reorganized, and the key points of the MASLD progression have been highlighted.
4.How lipid droplets are involved in the MASLD progression.
Reply: We sincerely thank the Reviewer for this precious comment. We are aware that Lipid droplets (LDs), once regarded as inert reservoirs for neutral lipids, are now recognized as dynamic organelles that play a central role in cellular metabolism and immune regulation. In the context of Metabolic dysfunction-associated steatotic liver disease (MASLD), the excessive accumulation of LDs within hepatocytes reflects a state of metabolic imbalance, primarily driven by insulin resistance and dysregulated lipid homeostasis. This lipid overload induces lipotoxic stress, which compromises mitochondrial integrity and promotes the generation of reactive oxygen species (ROS). These ROS act as signaling molecules that activate innate immune pathways, including the NLRP3 inflammasome and NF-κB signaling, leading to the release of pro-inflammatory cytokines such as TNF-α, IL-6, and IL-1β. Beyond hepatocytes, LDs are also abundant in immune cells like Kupffer cells and infiltrating macrophages, where they serve as platforms for the synthesis of bioactive lipid mediators, including prostaglandins and leukotrienes. These lipid-derived signals further amplify the inflammatory response and contribute to immune cell recruitment and activation. Moreover, LDs support the metabolic reprogramming of immune cells, fostering a pro-inflammatory phenotype associated with trained immunity. Collectively, these mechanisms position LDs as active participants in MASLD progression, bridging metabolic dysfunction with chronic hepatic inflammation and fibrogenesis.
All these features have been added, properly discussed, and supported (respectively in the description “traditional pathogenesis” and in the paragraph dedicated to the novel “immunometabolic pathways”).
5. Write few sentences of the legends of the Figure 1. To reflect that what is happening in traditional and trained immunity.
Reply: We thank the Reviewer for this notification. In the resubmitted version of our manuscript, a few sentences describing the main mechanisms of traditional, in contrast with trained immunity, driving the MASLD to MASH progression have been added in the figure legend (Figure 1).

Reviewer 3 Report
Comments and Suggestions for Authors
The manuscript entitled "From “traditional” to “trained” immunity: exploring the novel frontiers of immunopathogenesis in the progression of Meta-bolic Dysfunction-Associated Steatotic Liver Disease (MASLD) " is well prepared. Some minor revisions should be made.
- The background part should be shortened.
- A table or a figure of the comparison between “traditional” and “trained” immunity should be added. What are the differences between them?
- The following publications merit citation as additional references: Food Bioscience, 2025, 66:106195; Nutrients, 2023, 15(3):496. It is recommended that the author includes these in the references section
- The conclusion part should be one paragraph
Author Response
Reviewer 3
The manuscript entitled "From “traditional” to “trained” immunity: exploring the novel frontiers of immunopathogenesis in the progression of Metabolic Dysfunction-Associated Steatotic Liver Disease (MASLD) " is well prepared. Some minor revisions should be made.
- The background part should be shortened.
Reply: We sincerely thank the Reviewer for this suggestion. In the resubmitted and reorganized version of our manuscript, the background section has been properly shortened, and a specific per se paragraph has been dedicated to describing an overview of the transition from “classic” to “untraditional” MASLD pathogenesis.
2. A table or a figure of the comparison between “traditional” and “trained” immunity should be added. What are the differences between them?
Reply: We sincerely thank the Reviewer for this precious suggestion. According to this, a table (Table 1) 1 comparing “traditional immunity” with “trained immunity” response, reporting the most relevant differences, as well as the crucial specific potential implications in MASLD pathogenesis, has been added in the resubmitted version of our manuscript.
3. The following publications merit citation as additional references: Food Bioscience, 2025, 66:106195; Nutrients, 2023, 15(3):496. It is recommended that the author includes these in the references section
Reply: We thank the Reviewer for this notification. As suggested, in the resubmitted version of our manuscript, the proposed references have been properly added in the references section as additional references.
4. The conclusion part should be one paragraph
Reply: We thank the Reviewer for this suggestion. In the resubmitted version of our manuscript, the conclusion section has been reorganized by remarking on the key points of immune response sustaining the MASLD progression, and relative potential translational implications.

Reviewer 4 Report
Comments and Suggestions for Authors
The manuscript "From "traditional" to "trained" immunity: exploring the novel frontiers of immunopathogenesis in the progression of Metabolic Dysfunction-Associated Steatotic Liver Disease (MASLD)" summarizes the newest insights of MASLD transformation into deadly pathologies as steatohepatitis (MASH) or hepatocellular carcinoma due to immunometabolism dysfunction. The authors describe how the loss of immune tolerance and liver cell integrity leads to the failure of homeostatic processes, increasing the risk of chronic inflammation, autoimmune disorders, and the development of liver tumors. The recent evidence that indicates the possible role of innate immune cells in MASLD to enter into a "trained" state through epigenetic reprogramming and metabolic rewiring triggered by metabolic and environmental insults which finally leads to persistent inflammation accelerating disease progression (such as fibrosis) beyond traditional hepatic injury pathways are also reviewed.
The manuscript suggests that understanding trained immunity in MASLD can open new treatment avenues aimed at modulating innate immune memory to restrain chronic inflammation and liver damage.
The manuscript is of scientific interest, is well documented and points out the important aspects of how immune dysregulation can promote liver disease progression.
However, in my opinion it has some unnecessary information.
Major comments:
- the background section it would be sufficient to include a concise paragraph summarizing the global epidemiology of Metabolic dysfunction-Associated Steatotic Liver Disease (MASLD and its impact.
- Regarding section 2.1 "Liver as an Immunological Organ", the detailed general descriptions of hepatocyte polarity, sinusoidal endothelial cells, fenestrations, and microanatomical details can be minimized or omitted if they do not contribute directly to the novel insights of the manuscript. The focus should be on the immune functions of liver cells relevant to MASLD pathogenesis or immunological mechanisms pertinent to the study's specific aims.
For instance, the detailed structural descriptions like:
line 208-210: "Hepatocytes carry out around 70% of liver functions and act as a key barrier between sinusoidal blood and bile..."
hours
Line 218: "LSECs make up 50% of non-parenchymal liver cells and line the low-shear sinusoidal capillaries. Unlike typical capillaries, they possess fenestrations and lack a basement membrane, facilitating substrate exchange between blood and hepatocytes." And so on..
All these details could be replaced by a brief explanation of highlighting the liver's immune function and cell types involved in MASLD-related immunopathology, thereby sharpening the manuscript's focus and avoiding dilution by widely known background details.
The authors should aim to improve manuscript clarity and impact, to prevent readers from getting lost in excessive or unnecessary details.
Author Response
Reviewer 4
The manuscript "From "traditional" to "trained" immunity: exploring the novel frontiers of immunopathogenesis in the progression of Metabolic Dysfunction-Associated Steatotic Liver Disease (MASLD)" summarizes the newest insights of MASLD transformation into deadly pathologies as steatohepatitis (MASH) or hepatocellular carcinoma due to immunometabolism dysfunction. The authors describe how the loss of immune tolerance and liver cell integrity leads to the failure of homeostatic processes, increasing the risk of chronic inflammation, autoimmune disorders, and the development of liver tumors. The recent evidence that indicates the possible role of innate immune cells in MASLD to enter into a "trained" state through epigenetic reprogramming and metabolic rewiring triggered by metabolic and environmental insults which finally leads to persistent inflammation accelerating disease progression (such as fibrosis) beyond traditional hepatic injury pathways are also reviewed.
The manuscript suggests that understanding trained immunity in MASLD can open new treatment avenues aimed at modulating innate immune memory to restrain chronic inflammation and liver damage.
The manuscript is of scientific interest, is well documented, and points out the important aspects of how immune dysregulation can promote liver disease progression.
However, in my opinion it has some unnecessary information.
Major comments:
1.The background section it would be sufficient to include a concise paragraph summarizing the global epidemiology of Metabolic dysfunction-Associated Steatotic Liver Disease (MASLD and its impact.
Reply: We sincerely thank the Reviewer for this valuable suggestion. In the resubmitted version of our manuscript, the background section has been reorganized and, as suggested, a concise paragraph summarizing the global epidemiology of Metabolic Dysfunctional-Associated Steatotic Liver Disease (MASLD) and its impact, highlighting prevalence trends, and associated health risks has been added.
2. Regarding section 2.1 "Liver as an Immunological Organ", the detailed general descriptions of hepatocyte polarity, sinusoidal endothelial cells, fenestrations, and microanatomical details can be minimized or omitted if they do not contribute directly to the novel insights of the manuscript. The focus should be on the immune functions of liver cells relevant to MASLD pathogenesis or immunological mechanisms pertinent to the study's specific aims.
For instance, the detailed structural descriptions like:
- line 208-210: "Hepatocytes carry out around 70% of liver functions and act as a key barrier between sinusoidal blood and bile..."
- Line 218: "LSECs make up 50% of non-parenchymal liver cells and line the low-shear sinusoidal capillaries. Unlike typical capillaries, they possess fenestrations and lack a basement membrane, facilitating substrate exchange between blood and hepatocytes." And so on..
All these details could be replaced by a brief explanation of highlighting the liver's immune function and cell types involved in MASLD-related immunopathology, thereby sharpening the manuscript's focus and avoiding dilution by widely known background details. The authors should aim to improve manuscript clarity and impact, to prevent readers from getting lost in excessive or unnecessary details.
Reply: We sincerely thank the Reviewer for this valuable suggestion, and we completely agree with His/Her point of view. As proposed, in the resubmitted version of our manuscript, in the paragraph dedicated to the description of the liver as an immunological organ, we have omitted and/or minimized the microanatomical details and focused exclusively on features concerning the immune functions of liver cells relevant to MASLD pathogenesis, highlighting the relative immunological mechanisms. Overall, the clarity of the manuscript has been improved by avoiding redundancies of details not pertinent to the study’s specific aims.
